# Searching for Life, Mindful of Lyfe’s Possibilities

**DOI:** 10.3390/life12060783

**Published:** 2022-05-25

**Authors:** Michael L. Wong, Stuart Bartlett, Sihe Chen, Louisa Tierney

**Affiliations:** 1Earth & Planets Laboratory, Carnegie Institution for Science, Washington, DC 20015, USA; 2Division of Geological & Planetary Sciences, California Institute of Technology, Pasadena, CA 91125, USA; sjbart@caltech.edu (S.B.); sihechen@caltech.edu (S.C.); 3The Potomac School, Science Engineering & Research Center, McLean, VA 22101, USA; ltierney@potomacschool.org

**Keywords:** habitability, origin of life, astrobiology, biosignatures, genesity, lyfe

## Abstract

We are embarking on a new age of astrobiology, one in which numerous interplanetary missions and telescopes will be designed, built, and launched with the explicit goal of finding evidence for life beyond Earth. Such a profound aim warrants caution and responsibility when interpreting and disseminating results. Scientists must take care not to overstate (or over-imply) confidence in life detection when evidence is lacking, or only incremental advances have been made. Recently, there has been a call for the community to create standards of evidence for the detection and reporting of biosignatures. In this perspective, we wish to highlight a critical but often understated element to the discussion of biosignatures: Life detection studies are deeply entwined with and rely upon our (often preconceived) notions of what life is, the origins of life, and habitability. Where biosignatures are concerned, these three highly related questions are frequently relegated to a low priority, assumed to be already solved or irrelevant to the question of life detection. Therefore, our aim is to bring to the fore how these other major astrobiological frontiers are central to searching for life elsewhere and encourage astrobiologists to embrace the reality that all of these science questions are interrelated and must be furthered *together* rather than separately. Finally, in an effort to be more inclusive of life as we do not know it, we propose tentative criteria for a more general and expansive characterization of habitability that we call *genesity*.

## 1. How One Defines or Characterizes Life Is Critical to Logical and Lucid Biosignatures Research

There is an old chestnut, often promulgated around Astrobiology Science Conferences, that if you ask 100 astrobiologists for the definition of life, you will get 150 different answers. As of this writing, there is no definitive definition of life, nor is there a complete, universally agreed-upon theory of life [1,2]. However, the goal of biosignatures research is to find evidence for life in the universe. So how do we know what we are looking for?

Some contend that we do not need a definition of life in order to search for it successfully. Frequently, this argument goes, “we’ll know it when we see it”. However, history offers a cautionary tale. For many centuries, humans mistakenly attributed the symptoms of microbial infections to non-living factors, such as miasma (“bad air”). Only after the cell theory of life and germ theory of disease were developed in the 19th century did our modern picture of infection emerge. Similarly, a universal theory for life may need to be developed before we can appreciate the true diversity of living systems in the universe.

In scientific writing, it is often assumed that the authors and the reader share the same conception of life. However, this is far from guaranteed; in fact, in certain instances, it may not even be clear that the authors of a manuscript share the same conception of life with *one another*. Thus, when speaking about the science of life detection (biosignatures, biomarkers, etc.), astrobiologists must first make plain their framework for life. How one conceptualizes life is the foundation for all downstream astrobiological arguments, forming the framework in which the concepts of habitability and biosignatures gain specificity.

There are many different working definitions of life; our goal here is not to advocate for one over another. (Indeed, we would argue that, given our present state of ignorance, all definitions thus far are actually *tentative criteria* for life that must be updated as knowledge advances.) Rather, we advocate that researchers clearly define what mode they are working within when presenting evidence and arguments for (or against) “life” elsewhere. To first order, astrobiologists should make clear: Are they talking about detecting “life as we know it here on Earth, but in an extraterrestrial environment” or finding evidence for “a general characteristic of life”? Henceforth, we will follow [3]’s characterizations of life (for the former) and lyfe (for the latter) and put “life” in quotation marks when it is meant to be ambiguous.

Not being clear about one’s framework for “life” can lead to confusion, frustration, and logical inconsistencies when interpreting signals and conveying their meaning to others. For example, one might naïvely try to use a product of life (as we know it) to justify the detection of lyfe (as we do not) in an environment that perhaps cannot support life. Let us consider an extraterrestrial environment that is far beyond the physicochemical boundaries for life (e.g., Venus’s sulfuric acid cloud droplets [4]) where a specific expression of life on Earth is found (e.g., a metabolite manufactured by life). The argument that this signal is a sign of life suffers from a logical inconsistency: It is like finding a backgammon piece and using it to argue that chess is at play—they do not even use the same board. If a living system does exist in this extraterrestrial environment, it would almost certainly utilize different biochemistry from Earthly life, making it problematic to affiliate Earth-specific biosignatures with alien lyfe simply on the basis that they are signs of life on Earth.

Note that lyfe may well exist in this environment and may even be the source of the purported signal; however, evidence for such lyfe would require metrics that are *agnostic* to the specifics of life on Earth. Hence, environments that are outside the “habitable” parameter space with respect to Earthly life necessitate a discussion of lyfe and agnostic biosignatures at the outset. Such general descriptions for lyfe may not be detailed enough to predict specific biomolecules, but they may still predict more general classes of biosignatures—all of which can still be effective for the detection of Earthly life—that can be used to motivate the detection of truly alien lyfe. These agnostic approaches include homochirality, chemical disequilibria e.g., [5,6], elemental distributions [7], molecular complexity [8], and planetary complexity [9].

Thus, researchers should be careful to state the assumptions that they are making about life and elucidate the relationship of their proposed or purported biosignature to their characterization of life. This will allow both authors and readers to identify various caveats based on these assumptions when interpreting scientific results. Meanwhile, all astrobiologists—including and especially those interested in seeking biosignatures—should encourage and take into account the continued development of a general theory of life [1].

We wish to highlight the fact that there are many possibilities with respect to the correspondences between habitable environments and forms of lyfe. As shown in Figure 1, there are four general possibilities: (a) there are many types of habitable environments, and each is uniquely associated with its own form of lyfe; (b) there is only one type of habitable environment, but it can give rise to and support many forms of lyfe; (c) there are many potentially habitable environments but only a single form of lyfe (specifically, life) that will emerge in all of them; and (d) there are many potentially habitable environments and each has the potential to give rise to and support different forms of lyfe. We suggest that the present state of knowledge (lack of constraints) implies we have to accept (d) until we find evidence to the contrary since it is the most general of the four options. Of course, not *every* link between environments and lyfe will be viable due to the availability of fundamental resources in certain environments. Hence, reality may look more like (e), where any instance of lyfe can be supported by a range of environmental space, and any particular point in environmental space can support a wide array of lyfe. In any scenario where multiple forms of lyfe may exist in the universe—i.e., in all but scenario (c)—it will be critical to developing agnostic biosignatures techniques for identifying signs of any form of lyfe that might exist in a given environment.

## 2. A Better Understanding of the Origins of Life Is Important to Biosignatures Research

The detection of life may not be a single, instantaneous event. Rather, evidence collected over time and through multiple approaches may lead to higher (or lower) confidence that the hypothesis of life’s presence is true. Hence, there is a growing consensus in the community that a *probabilistic* framework for life detection must be developed [10,11].

In a Bayesian framework, the probability of the life hypothesis (*L*) being true, rather than the no-life hypothesis (*N*), given data collected (*D*) and the context of the environment (*C*), is:(1)$$P\left(L|D,C\right)=\frac{P(D,C|L)P(L|C)}{P(D|C,L)P(L|C)+P(D|C,N)P(N|C)}$$

As seen in Equation (1), the probability of abiogenesis given the environmental context, *P*(life|context), is especially important in a Bayesian approach to life detection. *P*(life|context) constitutes the “prior” in Bayes’ equation. Note that *P*(no life|context) is simply 1–*P*(life|context).

In other words, the likelihood of life emerging on the planet of interest plays an important role in determining the probability that a signal of interest is due to life. However, the question of origins is often swept under the rug, often deemed “beyond the scope of the study.” Yet, as Equation (1) shows, the probability of the emergence of life is *not* “beyond the scope” of biosignature detection but *inherently crucial* to it.

By way of example, a recent study applied a probabilistic framework to data collected by the Cassini mission and claimed that Enceladus’ methane was most probably due to life [12]. In this study, “life” refers to direct analogs of Earth’s prokaryotic methanogens. This study presented the likelihood of life, *P*(life|data, context), as a function of *P*(life|context). By varying *P*(life|context) from 0.05–0.95, the study showed that *P*(life|data, context) > *P*(no life|data, context) when *P*(life|context) ≥ 0.35. But what if *P*(life|context) ~ 10^−3^ or 10^−6^ or 10^−47^? The reality is that we simply do not know. We applaud this study’s use of a Bayesian framework that acknowledges *P*(life|context) as part of their analysis. However, the study also demonstrates how our general ignorance about the origins of life directly impacts the interpretation of potential biosignatures. The fact that *P*(life|context) remains highly unconstrained poses a fundamental concern for the interpretation of potential biosignatures in any situation where the detection of life is neither immediate nor obvious.

Moreover, a subtle but critical confounding issue arises whenever “context” is equated with “a presently habitable environment”, such as the oceans of icy worlds. In most cases (excepting those dealing with fossil biosignatures from an ancient environment), present-day habitability is the correct “context” for the term *P*(data|context, life), i.e., the probability that life in this environment *today* would produce the observations *now*. However, where the emergence of life is concerned, “context” must relate to conditions that could give rise to de novo life, a parameter space that may or may not be congruent with the conditions for “habitability” (in whatever way “habitability” is defined within the chosen framework for “life”; see Section 3). The “context” in *P*(life|context) implicitly includes the planet’s evolutionary history, which may be ascertainable (as in the case of Mars e.g., [13]) or challenging to determine (as in the case of exoplanets). The environmental parameters conducive to the emergence of life are hotly debated for Earthly life and wildly unconstrained for lyfe as we do not know it. Because most narratives for the emergence of life are based upon ancient terrestrial conditions e.g., [14,15,16,17], we have hardly begun to examine the full suite of conditions that could result in abiogenesis.

Bettering our understanding of life’s emergence not only brings us closer to penetrating the mystery of our own beginnings, it also affects our interpretation of potential biosignatures through constraining *P*(life|context). If, for example, an origins-of-life experiment of unprecedented scale, scope, and complexity is proposed, it should be undertaken in collaboration with the biosignatures community and be simultaneously regarded as a biosignatures project. Indeed, determining whether such an experiment succeeds or fails at creating de novo life would be one of the most instructive biosignatures exercises ever conceived. Even if no life emerges, “failure” in this regard would still be instructive by way of understanding the extent to which prebiotic chemistry (most of which has not yet been explored or even imagined) can create life-like features without fully becoming life [18].

Yet, instead of working side-by-side, the biosignatures and origins communities largely conduct their research separately. This is surely not helped by the fact that within the origins community itself, great divisions remain. Some of this internal friction can be attributed to the nebulous end goal of what “life” truly is [3,19], making the message of Section 1 equally germane to that domain. Nonetheless, we hope to encourage greater dialogue between biosignatures and origins researchers. While their astrobiological goals may seem to be literally worlds apart, they are actually two sides of the same coin.

## 3. Developing a More General Concept of Habitability (Genesity) Is Important to Biosignatures Research

The search for life often begins with habitable or once-habitable environments. Habitability is traditionally defined, to first order, by the presence of liquid water, hence the concept of the “habitable zone” [20]. When it comes to life as we know it, habitability is often characterized as the confluence of liquid water, free energy (chemical or solar), and the availability of the basic chemical ingredients for life (CHNOPS), bounded in parameter space by the physiological limits of life on Earth e.g., [21]. However, this is unlikely to be an adequate definition of habitability as it pertains to lyfe, which may use different free energy sources and chemical constituents. Furthermore, once lyfe emerges, habitability becomes a function of biological activity as well as geological, chemical, and physical forces. Habitability is thus a dynamic concept, such that a planet’s habitability can depend on various internal and external factors, as well as time.

### 3.1. Proposed Criteria for Genesity, a More General Concept of Habitability

Is there a general theory of habitability that can be extended beyond terrestrial surfaces and ocean worlds to include hydrocarbon worlds, super-Earths, sub-Neptunes, gas giants, and the great diversity of bio-techno-planetary environments that may arise? How should we define the concept of habitability to encapsulate these wider possibilities?

Given our state of knowledge (or ignorance) about the nature of lyfe in the universe, developing a general concept of habitability is as challenging as developing a universal theory of life. Here, we identify three potentially quantifiable characteristics that could contribute to a new metric for generalized habitability, which we call *genesity* (Figure 2):
**Energetic driving force** (EDF)A requisite set of one or more free energy gradients that provide a sufficient thermodynamic force for the fluxes necessary for starting, maintaining, and complexifying lyfe. Note the additional requirement that such gradients cannot be trivially dissipated by abiotic processes (in such a case, the abiotic channels would become the paths of least resistance).

2.**Informational driving force** (IDF)The environment must exhibit a threshold level of complexity such that there is a selection pressure for the information processing and learning behaviors that are at the core of the living state [22,23,24,25,26,27]. There is no physical or computational reason for a lyfeform to emerge or evolve in an environment that is trivially simple. The IDF highlights an important difference between habitability and genesity: a perfectly placid pool of organic broth may be an optimally habitable environment, but it would have a negligible IDF and, therefore, low genesity.

3.**Combinatorial diversity of components** (CD)A sufficiently diverse set of components (molecular or otherwise) is required for the emergence of prebiotic systems that can harness the EDF and find dynamical robustness in resolving the information gradients provided by the IDF. The minimal set of such components or their chemical identity is not yet known, but we can surmise that for a given environment, there is a minimal component set required for the emergence of the lowest levels of biological complexity.

Genesity may therefore be defined as a function of three parameters:*G* = *f*(EDF, IDF, CD).

For each of these tentative criteria for genesity, we imagine an “ideal” region for lyfe that is neither too high nor too low (Table 1), in accordance with the notion that lyfe is a complex emergent phenomenon that exists “between order and chaos” [28,29].

We choose a new term, genesity because we believe this concept conveys something deeper about an environment than traditional notions of habitability. The criteria for habitability describe environmental features that allow life as we know it to survive. Essentially, habitability asks the question, “To what degree can this environment support Earthly biological activity now?” Genesity, on the other hand, encompasses survival but also describes an environment’s potential for the origin (genesis) and evolution (generation of novelty) of lyfe. Genesity asks the question, “To what degree can this environment originate and support the open-ended development of biology over evolutionary time?” Because these are different queries, introducing a new term preserves the traditional, and somewhat more specific, the meaning of “habitability” for situations in which it does indeed apply. Because genesity is based on abstract features of the environment, it should serve as a suitable starting point in our search for both life and lyfe.

We define an environment with a relatively high genesity value as a *genial* environment. A genial environment is necessary for the emergence and evolution of lyfe, but this outcome is not guaranteed due to the chaotic and indeterminate nature of complex evolving systems. This is similar to the concept of habitability, which provides necessary but insufficient criteria for the existence of life in an environment (hence the common caution in astrobiological discourse that *habitable* does not imply *inhabited*). Genesity may also describe an environment’s potential to host complex evolving systems at large, regardless of whether they truly qualify as lyfe [30,31]. We consider the genesity of one such sublyfe system, namely evolution within the environment of cyberspace, in Section 3.3.

The necessity of information processing phenomena for abiogenesis is a matter of ongoing debate in the astrobiology field. Some would contend that the dividing line between lyfe and non-lyfe lies before the point at which information processing plays a key role. However, our own position is that any form of sub-lyfe lacking any kind of learning or evolutionary potential does not fulfill the sufficient criteria for lyfe and is thus likely a “dead end.” Remote detection of such sub-lyfe (e.g., a proto-metabolism, molecular homochirality, chemical pattern formation, autocatalysis, or oscillations) would, of course, be astrobiologically significant. Such phenomena are likely crucial for lyfe and abiogenesis in a broad sense, but strict subsets of them do not fulfill the definition of lyfe. Although this represents the current perspective of the authors, we welcome and appreciate the entire spectrum of opinions on this important question.

### 3.2. Genesity Parameter Evolution and Co-Dependence on Planetary Context

Our three proposed criteria for genesity are interdependent upon one another and the emergence and evolution of lyfe itself. For instance, a key feature of a high-IDF environment is the presence of local minima in entropy, which is itself related to free energy gradients that could be harnessed by (proto-)lyfe forms. Once it has established a foothold in the environment, lyfe will introduce new niches and selection pressures, increasing environmental complexity and hence the IDF. Not only will lyfe impact the evolution of the metrics over time, but it may also redefine the ideal “sweet spot” for further evolution.

Additionally, there are likely different “ideal” regions for the *emergence* of lyfe and the open-ended *evolution* of lyfe: for example, the modern biosphere may have increased the environmental complexity of Earth such that it is unlikely for de novo life to arise spontaneously. On the other hand, it is possible that life on Earth has improved the chances for new kinds of artificial and digital lyfe to arise (see Section 3.3). This in silico and dataomic lyfe would not be possible without extant life (humans) building its required CD (e.g., computers) and providing high enough levels of EDF and IDF. In Figure 3, we show various hypothetical evolutionary trajectories for our genesity parameters of astrobiologically relevant planetary bodies.

On Earth (Figure 3a), the EDF begins too high for life during the planetary accretion and magma ocean phases, then quickly drops to a level ideal for the emergence of life. It rises during the Great Oxidation Event, which enables aerobic respiration and the diversity of more complex life e.g., [32,33]. The IDF rises after the emergence of life and, due to life’s presence, enters a regime where it is ideal for evolution but not further (organic) emergence events. At the beginning of the rise of oxygen, the spatiotemporal variability of oceanic O_2_ may have boosted the IDF and promoted evolutionary innovations in metazoan life forms [34]. The recent rise of human civilization and the dataome provides the EDF, IDF, and CD for the emergence and rapid evolution of artificial life forms (Alife). This increase in genesity for life may also represent a concomitant detriment to the environment’s genesity for organic life by way of anthropogenic climate change and a human-driven “sixth extinction” [35].

The evolutions of Mars (Figure 3b) and Venus (Figure 3c) offer alternative trajectories for terrestrial planets. Mars’s Noachian period may have been similar to the early Earth and have supported the emergence of life [36,37,38]. However, due to atmospheric and water loss during the Hesperian period, EDF, IDF, and CD all decay to values likely too low to support the global proliferation of life in the present day. Ancient Venus may have been habitable for life as we know it, but at some point, this clement Venus would have transitioned into its present-day runaway greenhouse state [39]. During that transition, the IDF and CD of the planet’s surface environment would have dropped below the threshold values for complexifying and maintaining lyfe. Note that the trajectories of these metrics for the cloud layers may be different; for example, in the cloud layers the radiation environment provides an EDF comparable to that of Earth’s surface [40].

In the outer Solar System, the icy satellites of Jupiter and Saturn provide different kinds of astrobiological targets. On Europa (Figure 3d), ice radiolysis and hydrothermal activity may contribute a redox gradient (EDF) capable of supporting microbial and even macrofauna analogs in this icy moon’s subsurface ocean [41,42,43,44,45,46]. However, its abiotic IDF is unlikely to be as high as that of Earth, but it may increase if life emerges and creates new biological feedback. Photochemistry in Titan’s (Figure 3e) thick N_2_–CH_4_ atmosphere samples a wide range of chemical space (high CD e.g., [47]. Titan’s IDF is fairly high due to the numerous environments it supports: seas, lakes, rivers, sand dunes, cryovolcanism, etc. However, it has a relatively low EDF due to its distance from the Sun and its low temperature (~94 K). Titan’s surface is a representative example of a location where the traditional concept of habitability breaks down (due to its lack of liquid water), but the concept of genesity is still informative.

Many planetary environments may be unsuitable for any form of lyfe. The atmospheres of hot Jupiters (Figure 3f), for example, may have EDFs in excess of that conducive to lyfe. Simultaneously, these planets may have too low a CD for lyfe because they are unable to support the existence of complex molecules in their atmospheres, which are thermodynamically unstable at >2000 K temperatures. On the other hand, a hypothetical “chaotic world” (Figure 3g) may experience wild environmental swings, either externally or internally induced. Although one origin of lyfe may have occurred early on in such an unpredictable world, the biosphere could be demolished when IDF or EDF becomes too high or low.

At the other end of the spectrum, there is no reason to believe that Earth represents the pinnacle of planetary genesity. For a hypothetical superhabitable environment (Figure 3h), EDF, IDF, and CD would promote the emergence of lyfe early in the planet’s history, then quickly evolve to a near-optimal state for the evolution and complexification of lyfe, maintaining that state through a network of planetary-scale homeostatic feedbacks. Such long-term homeostasis could be achieved through the onset of “planetary intelligence” at the pre- and post-civilization levels [48,49]. There are many potential superhabitable systems that Figure 3h could describe. Here we propose one set of planetological parameters that could satisfy a superhabitable state: First, with regard to EDF, we can imagine a planetary trajectory where EDF increases in a similar manner to the Great Oxygenation Event on Earth but without reliance on the evolution of oxygenic photosynthesis. Perhaps a more UV-active star would create abundant O_2_ via photochemical reactions, e.g., H_2_O or CO_2_ photolysis e.g., [50,51,52], enabling the evolution of more complex forms of life earlier in the planet’s history [53]. At the same time, this hypothetical world could generate a large flux of geochemical reductants if it has vigorous interior dynamics resulting in plate tectonics and magmatism. Such activity would result in copious volcanism, serpentinization, and other hydrothermal activity that emits ample amounts of reduced gasses, e.g., H_2_, H_2_S, Fe^2+^, or NH_3_. Not only would this robust geologic activity contribute to a high EDF but also to high CD and IDF by creating rich geochemical settings and time-varying environments, promoting profuse ecological diversity. Such a world may contain a superset of Earth’s geochemical environments, including biomes our minds are unable to comprehend yet, contributing even further to its IDF.

The process of organisms learning environmental features itself feeds back to the IDF: As living systems build more accurate models of their environments, those environments become modified away from the configurations in which the initial learning processes took place. Assuming again that the combinatorics of components permit, living systems would be expected to expand their predictive capacities further due to the enhanced free energy gains that would be yielded from such abilities. In addition to modifying the external environment, such systems would experience a selection pressure for the modeling of the biological agents as well, also contributing to increasing IDF values (as well as increasing the upper limit of the ideal IDF window) over time.

A deep philosophical question exists in this realm: What are the necessary and sufficient characteristics required for such open-ended learning and environmental modifications (as opposed to saturation of learning and emergence over time, as seen in the non-living world)? This is explored in the field of “open-ended evolution” e.g., [54,55,56,57,58,59,60,61,62]. The nature of open-ended learning and evolution is a deeply challenging and open area. It is entirely possible that learning or evolution can proceed to a certain extent and then experience a “saturation of complexity,” reach some kind of configurational upper bound, or get indefinitely stuck in an evolutionary local optimum. In such a case, for a planet with a relatively low upper bound of this kind, remote detection of its biosphere would be particularly challenging since it may be indistinguishable from its abiotic environs [9]. However, open-ended complexification is not a necessary component of the genesity framework: limited or saturating complexification of a biosphere is entirely compatible with the concept. Indeed, future observations of a range of biospheric complexity levels will be crucial to understanding our place in the universe.

We also wish to highlight the possibility of a binary distribution of biospheric complexity in the universe due to the Gaian bottleneck effect e.g., [63,64,65]. If there is a threshold level of complexity required for a biosphere to achieve the requisite homeostasis that guarantees long-term viability, then those biospheres that fall short of passing through the Gaian bottleneck would remain at low levels of complexity or simply disappear, and those that pass through may universally experience open-ended complexification. This would result in a binary distribution of biospheric complexities (with a heavy tail for the higher group due to open-endedness).

### 3.3. The Genesity of Cyberspace and the Dataome

In social networks—in particular, cyberspace—where data are well documented and there are fewer thermodynamic constraints than in the physical world, we can observe a range of complex and emergent phenomena [66], which can also be viewed as another form of lyfe under this particular type of environment see also [67]. Therefore, our proposed metrics for genesity may also apply to the realm of artificial and digital lyfe forms, as well as to technology and the dataome [68].

Perhaps the most salient form of a lyfe-like entity in cyberspace could be described as a meme or a popular application. The replication advantage for such an entity is obtained when people spend more time engaging with it and give kudos to it (e.g., likes). Hence, the free energy analog, in this case, can be defined to be the associated attention and time commitment by internet users. Like genes, memes experience a selection pressure: those that allow people to express more information within a more compact expression will go viral faster. A key driving force for the memes’ propagation is the information gradient with respect to existing memes, i.e., people without knowledge of the memes will start to engage with them when they offer greater novelty or higher information density. When a meme has depleted such an information gradient (everyone has seen it, and it becomes “old news”), its replication will slow, and the next generation of memes will become popular. However, it is still possible (indeed likely) that components of the early memes will remain in the evolutionary process (the next generation is likely to contain elements of the previous one, having exchanged content through various mechanisms).

In the cyberspace environment, combinatorial diversity includes the technologies involved, the types of communication channels available, and the groups of users that have access to the internet. With a higher combinatorial diversity, more complex memes are made possible. For example, when internet bandwidth and storage devices improved significantly over the past decade, more GIF memes with higher information density emerged, while primarily word memes were prevalent before that.

Treating cyberspace as a special form of environment allows us to observe a special case of emergence and evolution in which replication, mutation, and selection take on different forms to the biosphere, and potentially new forms of lyfe are arising before our own eyes. A rudimentary understanding of evolution in this realm could have significant consequences for a more general and agnostic classification scheme for biological learning beyond Earth.

### 3.4. Toward the Quantification of Gensity Parameters for Biosignature Science

The traditional view of habitability (liquid water, energy sources, presence of CHNOPS) is attractive due to its ease of use; it is relatively straightforward to seek these attributes of a planetary environment using present-day and near-future technology. On the other hand, genesity may be significantly more difficult to put into use immediately. For instance, it would be difficult to ascertain the CD or IDF of a distant extrasolar world without detailed knowledge of its surface features and composition—capabilities just beyond the reach of today’s technology. Hence, at present, robustly determining signs of traditional habitability may represent the state of the art. Nonetheless, we believe it is important to introduce a theoretical framework for genesity because: (1) its greater abstractness makes it more widely suitable for considerations of lyfe as we do not know it; (2) it may reveal deeper insights into the connections between conditions suitable for the emergence of lyfe and conditions that support the maintenance and evolution of lyfe; (3) it may incentivize the development of new tools for quantifying EDF, IDF, and CD.

In this contribution, we have presented our criteria for genesity as qualitative sketches, but they all have promising avenues for rigorous quantification. For example, IDF could be estimated using the framework of statistical complexity and epsilon machine reconstruction from complexity science [28,69,70], in addition to the idea of requisite variety [71], i.e., that in order to exhibit homeostasis, systems must somehow implement a model of the sources of perturbations (the environment) against which homeostasis is being leveraged. The statistical complexity of a system is computed by analyzing time series measurements from that system and deducing the most compact and predictive finite state machine model for the variations in the time series. We can surmise that environments of very low statistical complexity do not provide any form of selection pressure or advantage to systems that can perform information processing (lyfe). Thus, in such environments, any free energy gradients would either be dissipated abiotically or become kinetic bottlenecks. In contrast, in environments where free energy gradients exist but can only be harnessed once a minimal level of information processing emerges, we would expect to see such biological emergence, assuming the combinatorics and availability of components (CD) allow for such.

At the same time, CD may be determined via molecular assembly theory, more specifically the molecular assembly index (MA), which provides a scale to measure molecular complexity (the higher the MA, the more complex the molecule is). By analyzing molecular assembly pathways (the steps required to form the molecule from its basic building blocks), MA illustrates the smallest number of steps required for molecule formation [8]. Using mass spectrometry, the mass-to-charge ratio of sample fragments can be measured, creating fragmentation spectra. The diversity of the fragmentation spectra correlates with MA; hence, such spectra can be used to infer the MA of the parent molecule. Furthermore, molecular assembly can be applied to a diverse array of samples through mass spectrometry and has the potential to be utilized on future missions to assess the molecular complexity of an environment. Hence, one could estimate CD by analyzing the distributions of MA values for a given planetary environment. In addition to the molecular assembly method, chemometric fingerprinting could also be utilized to quantify CD. Chemometric fingerprinting is a method in life detection in which many different DNA strands are projected onto a sample. The diversity of the strands that binds to the sample can be used as a proxy for the sample’s chemical complexity [72].

Finally, EDF may be determined via a combination of chemical disequilibria and the stellar input free energy. The atmospheric chemical disequilibrium, for instance, is a measure of the free energy that could be utilized by lyfe through chemotrophic metabolisms that involve atmospheric gasses e.g., [5]. Aqueous chemical disequilibria have been described for ocean worlds e.g., [73].

Stellar irradiation also provides a planetary system with free energy that can be harnessed by biological systems. Where genesity is concerned, it may be of interest to consider not just the *total* free energy input rate but also *local* gradients of free energy input across regions of a planetary surface. We would expect the origin of lyfe only when biotic processes are more favorable in terms of harvesting free energy compared to non-biotic processes. Processing information, in any form, could lead to increased information entropy and affect local free energy gradients. In the absence of a local free energy gradient, the emergence of lyfe would not be permissible.

If we consider two regions of the planet’s surface with different albedo values, their difference in free energy flux can be considered as a local free energy gradient. While the absolute free energy influx peaks close to the host star’s surface and decreases with orbital distance, it can be shown that this fractional difference increases monotonically with distance from the star. Thus, we may find an orbital distance where the absolute free energy influx and the local free energy gradient values compromise to create a high-genesity environment, where the system is both active enough to maintain competition and promote learning systems. We hypothesize that, over long timescales, a planetary biosphere may create new local free energy gradients through the introduction of new biomes and via its impacts on biogeochemical cycles and climate feedback, thereby enhancing the complexity and hence the genesity of the environment.

Exactly how stellar input free energy gradients may impact EDF and IDF, especially in the presence of a biosphere, is a matter that we leave for future work. This question is complicated by the fact that, for any given planet, the stellar input free energy will be determined not just by the planet’s orbital distance but also by its atmosphere. While the addition of an atmosphere reduces the free energy influx to the planet’s surface, the maximum position of free energy flux moves outwards from the star. This problem awaits further investigation with more detailed radiative transfer modeling.

## 4. Concluding Remarks

The NASA-sponsored “NfoLD/NExSS Standards of Evidence for Life Detection Community Workshop” laid out a general framework for rigorous biosignature assessment: (1) the detection of a signal; (2) the identification of that signal; (3) an assessment of abiotic sources for that signal; (4) an assessment of biological sources for that signal; and (5) independent lines of evidence to support the hypothesis. Steps 1 and 2 deal with the detection of a signal of interest, while steps 3–5 serve to interpret that signal in the context of its environment, our understanding of life, and subsequent data [74,75].

In this perspective, we have highlighted how Step 4 (phrased as the question, “Is it likely that life would produce this expression in this environment?”) is highly contingent upon our understanding of what life is, how it emerges, and what it means for an environment to be habitable. In other words, answering fundamental questions about the nature of life impacts the quality of biosignature science directly by improving the interpretation of signals of interest. Astrobiology is highly interdisciplinary: Steps 1–3 rely openly on the foundations of geology, atmospheric science, planetary science, astronomy, etc. We advocate that in approaching Step 4, biosignatures researchers form similar ties with the origins-of-life community and those who engage in developing a universal theory of life. The quest for life in the universe should be tackled hand-in-hand with the study of the nature and emergence of life.

However, beyond the interpretation of signals, these sister pillars of astrobiological inquiry impact the whole project of searching for life by guiding our overall approach to looking for signals of interest. Thus, we conclude this perspective by advocating for an open-minded approach to life in the universe. If we confine ourselves to defining life exactly as we know it on Earth instead of taking a general characterization of the living state, we may bias ourselves toward looking for certain Earth-specific biosignatures that are not fundamental or universal to lyfe. Operating under definitions that are too narrow may blind us to detections hiding in plain sight, presenting the possibility of confirmation bias (or the so-called “streetlight effect”). Hence, we consider the development of agnostic biosignatures e.g., [8,9,72,76] critical to the future of lyfe detection.

Additionally, if we confine ourselves to one class of origins theory, we risk neglecting worlds where living systems could emerge in a different way. For example, operating in the paradigm that “warm, little ponds” exposed to atmosphere and sunlight were crucial to the onset of life would bias searches against ice-covered ocean worlds. To us, this stance seems premature given present uncertainties regarding geochemical environments suitable for abiogenesis. Even if “warm, little ponds” resulted in life here on Earth, this is no reason to discount the possibility that lyfe can emerge in various ways from various substrates in various milieus.

Similarly, if we confine ourselves to thinking about habitability as it is thought of on Earth, we may focus our search with too limited a scope [77]. In a Bayesian framework, our prior for lyfe in uninhabitable environments may implicitly be set so low that we fail to gather the relevant data or identify the relevant basis vectors for detecting it. Our planet is but one example of a living world—one particle on one path in an ensemble of evolutionary trajectories. A new, broader definition of habitability is needed for prioritizing the search for living systems in the universe. Here, we have proposed the concept of genesity based on three abstract environmental features that describe an environment’s ability to support the emergence and evolution of life: energetic driving force, informational driving force, and combinatorial diversity. Adopting the concept of genesity may embolden us to search for lyfe in places where life cannot exist.

To be clear, we are not discouraging research into signs of life (rather than lyfe), the origin of life (rather than lyfe), or habitability (rather than genesity). With limited time and resources, astrobiologists cannot be faulted for the relatively risk-averse position of looking for a second genesis of the single instance of life that we know to exist thus far. Our perspective is simply that a broader, more inclusive theoretical framing of lyfe has the potential to strengthen the astrobiological community and reshape our questions in ways that will eventually teach us more about our place in the universe.

We are embarking on a new age of astrobiology, and we must approach it *together* with humility and open-mindedness. This applies both to each other’s work and the greatest teacher of all: whatever is out there.

## Figures and Tables

**Figure 1 life-12-00783-f001:**
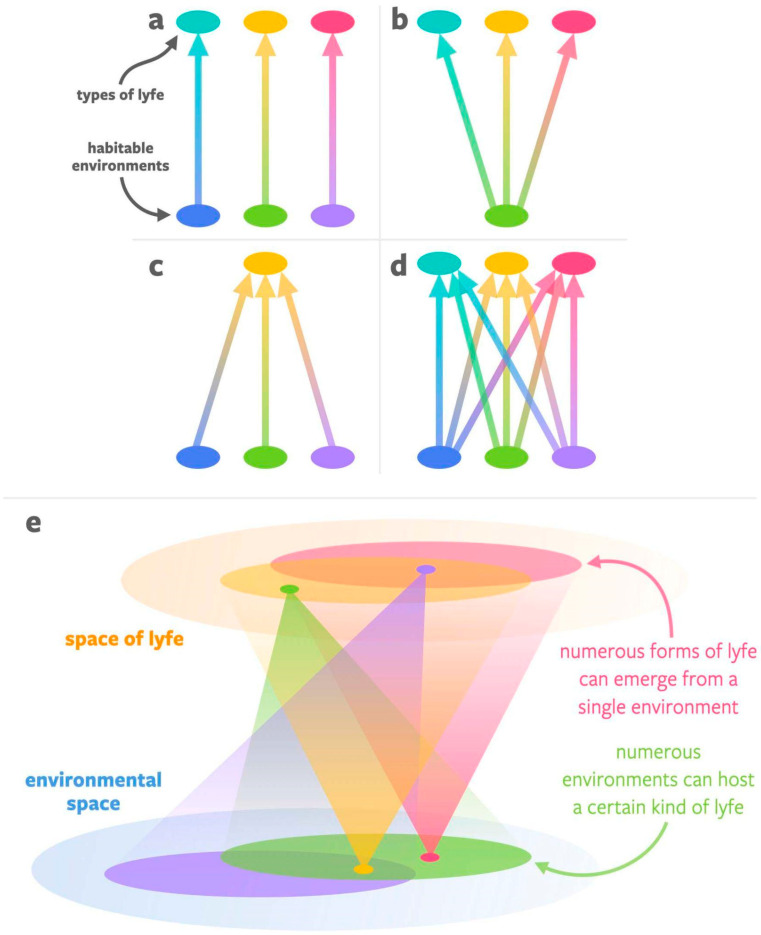
Four possibilities for the relationship between habitable environments (bottom of each subfigure) and types of lyfe (top of each subfigure). (**a**) A one-to-one correspondence: every habitable environment produces a unique form of lyfe. (**b**) A one-to-many correspondence: there is just one single class of habitable environment, but it is capable of generating and supporting myriad different kinds of lyfe forms. (**c**) A many-to-one correspondence: there are many different classes of habitable environments, but only one kind of lyfe (life) in the universe. (**d**) A many-to-many correspondence: there are many different classes of habitable environments, each of which is capable of generating and supporting myriad different kinds of lyfe forms. (**e**) A many-to-many correspondence but not every environment is capable of supporting every form of lyfe. If scenario (**e**) represents reality, then on other worlds—even Earth-like worlds—lyfe as we do not know it may emerge, and the development of agnostic biosignatures will be key to the search for life in the universe.

**Figure 2 life-12-00783-f002:**
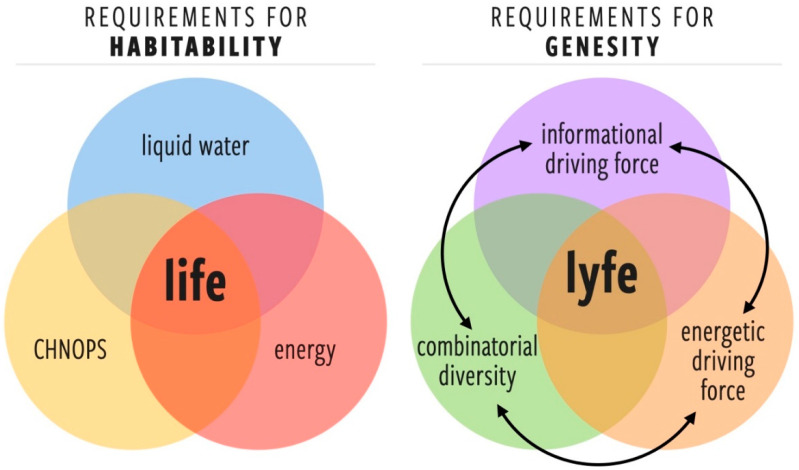
(**left**) The classic Venn diagram for habitability as it is traditionally defined for life and (**right**) a new Venn diagram highlighting the important factors for genesity as we define it for lyfe. The arrows between the factors contributing to genesity highlight that they are each functions of one another and that feedback exists between them (see Section 3.2).

**Figure 3 life-12-00783-f003:**
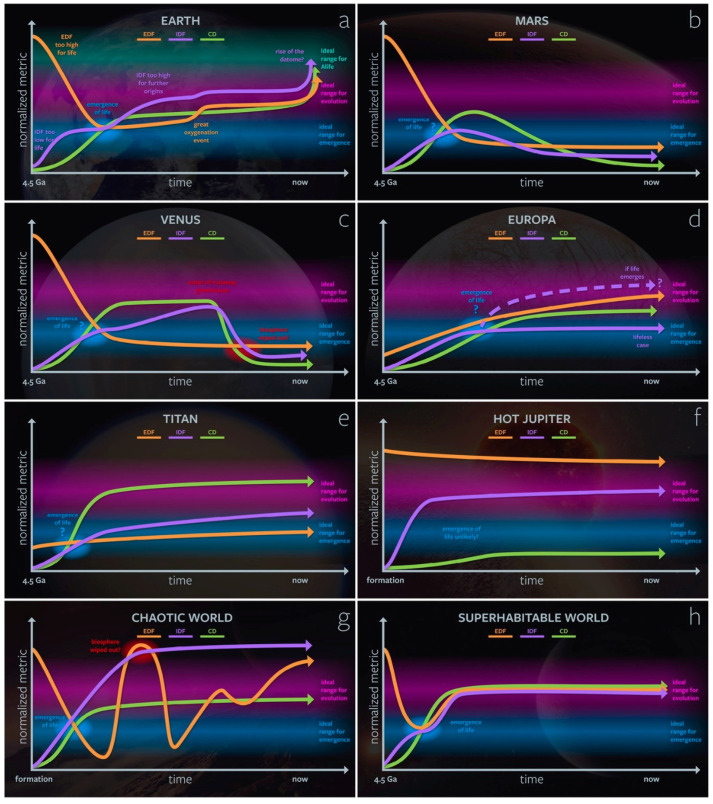
Schematic representations of EDF, IDF, and CD over time for different astrobiologically relevant worlds: (**a**) Earth, (**b**) Mars, (**c**) Venus, (**d**) Europa, (**e**) Titan, (**f**) a hot Jupiter, (**g**) a hypothetical “chaotic world,” (**h**) a hypothetical “superhabitable” world. Only certain major events are plotted for simplicity; in reality, these curves would be far more time-varying than shown here. We assume here that, for all three parameters, the ideal range for emergence is less than that for evolution/complexification. Importantly, the reader should note that these diagrams are all hypothetical; a robust calculation of these metrics over time has not yet been done.

**Table 1 life-12-00783-t001:** When any of the parameters for genesity is too high or too low, the environment is not conducive to lyfe.

Parameter	Too Low	Too High
*Energetic driving force*	Unable to supply work needed to maintain complexity, growth, and innovation	Turbulence; immediately overcomes kinetic barriers, producing chaotic behavior
*Informational driving force*	No incentive for evolution; no need for a living system	Prohibitively complex environment requiring an impossibly sophisticated first, putative learning system
*Combinatorial diversity*	Set of available components cannot produce systems above biological complexity threshold	Component configuration space too random/dispersed for biological emergence

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
