# Peer review of "Searching for Life, Mindful of Lyfe’s Possibilities"

_life, 2022, doi:10.3390/life12060783_

Round 1

Reviewer 1 Report

This is an interesting and thoughtful paper.  It is well written and meticulous in preparation, and I recommend publication in Life.  I have some comments that I hope the authors will find to be useful:

Major comments:

The fundamental principle of Bayesian probabilities is to update prior information with new data.  Bayesian techniques (albeit, fashionable) are of little utility when there's minimal information in the prior.  Do we really have a value for P(L|C)??  I think the paper needs a better justification of how a sensible value can be assigned to P(L|C).

line 211 "learning behaviors that are at the core of the living state".  I am dubious that information processing and learning behavior are intrinsic to lyfe.  I see no reason why a primitive lyfe form could not develop independent of IDF considerations.  It might not evolve, and it might be a dead end, but it could exist.  And if we detected it, that would be a huge scientific advance.  Right?

Minor comments:

line 63, please explicitly name the metabolite.  Are you referring to H2SO4?

Figure 3 is hard to discern, I can't read the fonts against the background.  The contrast needs to be improved.

line 276 "complexifying and maintaining lyfe"  >> is complexifying necessary?  And is "complexifying" a valid word in English?  (I like your terms lyfe and genesity, but be careful with terminology, especially because some readers may not be proficient in English).

line 332 "open ended learning"  >>  why would that necessarily occur?

Sec. 3.3  Are you saying that internet memes are lyfe?  Or just a close analogy?  Please be clear and explicit. 

Author Response

Please see the attached response to Reviewer 1's comments.

Reviewer 2 Report

General Comments

This paper, almost philosophical in nature, deals with the difficult issue of implicit assumptions underlying the whole astrobiology field, and aims to make these assumptions explicit. It also proposes a first framework to extend the validity of the astrobiology field, framework designed to become more quantitative when measurements and models have further progressed. I am not qualified enough to know if the authors succeeded in their attempt, but the paper raises important epistemological questions and has at least the merit to formulate them more clearly. I therefore recommend its publication, at least to spur further debate in the scientific community, once the comments below have been addressed by the authors.

Specific Comments

  • l.55: although not coined by the authors, I find the "lyfe" vs. "life" naming convention exceedingly confusing, especially when speaking (homophonous words). Since it is still an emerging field, nomenclature is not yet fixed, and very important to settle early on. I suggest that the authors use, in lieu of "lyfe", a more different word from life or scare-quoted ``life''.
  • l.123-145: I think that most astrobiologists are victims, consciously or not, of the "streetlight effect", when you are looking for your lost keys below the streetlight because (and only because) it is the only lit place. In this case, they stay focused on "organic chemistry in liquid water" because everything else is a mere conjecture and therefore a "dark" place. Since there is only a limited amount of resources can that be allocated to look for lyfe/life, they naturally stick to what has worked at least once somewhere (that is, here on Earth) and cannot be blamed for that risk-adverse position. I suspected the above discussion can be summarized in a Bayesian framework, and suggest that the authors do so here.
  • Section 3: I suggest that the feedbacks between EDF, IDF and CD be discussed a little bit in this section. For example, a key parameter to allow for the build-up of IDF is the possibility to sustain local minima in entropy, which is highly related to the (Helmholtz) free energy gradients that can be harnessed by (proto-)life forms. CD can also be understood as a mere substrate for information build-up and energy harnessing, so would seem less fundamental than (or at least different from) EDF and IDF. A figure with captioned positive/negative feedbacks arrows between EDF, IDF and CD might be helpful here.
  • l.230-232: In my opinion, it is even "worse" than that. Habitability is classically defined and understood as "To what degree this environment can support _Earthly_ biological activity, even including us humans?". It is even obvious in the etymology of the "habitability" word itself.
  • l.253-254: there is also the opposite possibility that life on Earth has improved the chances for (other kinds of) lyfe to arise, as discussed by the authors themselves in Sec. 3.3: in silico lyfe would not be possible without extant life (humans) building its required CD (computers) and bootstraping its EDF and IDF.
  • l.425-449: the typical time and length scales associated with the free energy gradients are of paramount importance to assess if they can be harnessed by lyfe or proto-lyfe. For example, on Earth, planetary-scale free energy gradients (and entropy creation through dissipative processes) are ruled by abiotic processes, mainly the climatic and oceanic heat engines. Competition between byotic (as pertaining to lyfe, whereas biotic pertains to life) and abyotic processes is to be assessed over a wide range of temporal and spatial scales, among other possible parameters. Here also, quantitative local entropy budgets should be estimated first.

Author Response

Please see the attached response to Reviewer 2's comments.
